# Evaluating the Memory Enhancing Effects of *Angelica gigas* in Mouse Models of Mild Cognitive Impairments

**DOI:** 10.3390/nu12010097

**Published:** 2019-12-30

**Authors:** Minsang Kim, Minah Song, Hee-Jin Oh, Jin Hui, Woori Bae, Jihwan Shin, Sang-Dock Ji, Young Ho Koh, Joo Won Suh, Hyunwoo Park, Sungho Maeng

**Affiliations:** 1Graduate School of Interdisciplinary Program of Biomodulation Collage of Natural Science, Myongji University, Yongin 17058, Korea; kms0177@naver.com; 2Graduate School of East-West Medical Science, Kyung Hee University, Yongin 17104, Korea; bungu20@naver.com (M.S.); julia110892@naver.com (H.-J.O.); wrbae0328@naver.com (W.B.); junikk0@naver.com (J.S.); 3Center for Nutraceutical and Pharmaceutical Materials, Myongji University, Yongin 17058, Korea; kimhwi83@gmail.com (J.H.); jwsuh@mju.ac.kr (J.W.S.); 4Department of Agricultural Biology, National Academy of Agricultural Science, Rural Development Administration, Wanju-gun, Jeollabuk-do 55365, Korea; sdji11@korea.kr; 5ILSONG Institute of Life Science, Hallym University, Anyang 14066, Korea; kohyh@hallym.ac.kr; 6Department of Bio-Medical Gerontology, Hallym University Graduate School, Chuncheon 24252, Korea; 7Health Park Co., Ltd., #2502, Gangnam-dae-Ro 305, Sucho-gu, Seoul 06628, Korea

**Keywords:** *Angelica gigas*, mild cognitive impairment, traumatic brain injury, chronic mild stress

## Abstract

(1) Background: By 2050, it is estimated that 130 million people will be diagnosed with dementia, and currently approved medicines only slow the progression. So preventive intervention is important to treat dementia. Mild cognitive impairment is a condition characterized by some deterioration in cognitive function and increased risk of progressing to dementia. Therefore, the treatment of mild cognitive impairment (MCI) is a possible way to prevent dementia. *Angelica gigas* reduces neuroinflammation, improves circulation, and inhibits cholinesterase, which can be effective in the prevention of Alzheimer’s disease and vascular dementia and the progression of mild cognitive impairment. (2) Methods: *Angelica gigas* (AG) extract 1 mg/kg was administered to mildly cognitive impaired mice, models based on mild traumatic brain injury and chronic mild stress. Then, spatial, working, and object recognition and fear memory were measured. (3) Result: *Angelica gigas* improved spatial learning, working memory, and suppressed fear memory in the mild traumatic brain injury model. It also improved spatial learning and suppressed cued fear memory in the chronic mild stress model animals. (4) Conclusions: *Angelica gigas* can improve cognitive symptoms in mild cognitive impairment model mice.

## 1. Introduction

Mild cognitive impairment (MCI) refers to a condition in which one’s cognitive function is lower than that of normal peers but not considered dementia [1,2]. However, patients with MCI have an approximately 50% chance of developing Alzheimer’s disease within five years [3]. Dementia is not curable. Therefore, it is important to prevent MCI from progressing into dementia. However, there is no established treatment that prevents the progression of MCI to dementia [4].

Animal models are needed to study MCI and develop therapeutics. An appropriate MCI model may have symptoms aggravating with age, but with only subtle memory impairment [5]. Animal models that meet these criteria include middle-aged rodents and transgenic mice that overexpress A β at an early stage before the dementia onset [4]. Spontaneously hypertensive rats (SHRs) appear to be appropriate as MCI models for vascular dementia, since hypertensive astrogliosis, cytoskeleton breakdown, hippocampal atrophy, and cholinergic deficit prematurely appear prematurely in this animal [6,7,8]. In contrast, drug-induced memory impairment models (such as those using scopolamine, NMDA blockers, and benzodiazepines) are not appropriate because they do not represent the various aspects of MCI [5].

Traumatic brain injury (TBI) is one of the most common brain injuries that causes a progressive decline of memory and cognition [9]. Unlike severe TBI, moderate to minimal TBI tends to be overlooked. However, even mild TBI can cause gradual amnesia, altered executive function, concentration disorders, depression, apathy, and anxiety [8,10,11]. In particular, repetitive head injuries, such as those caused by collision sports or motor vehicle accidents, are known to cause dementia [12]. Animal models of TBI show a decrease in cognitive function that correlates to the extent of injury, the number of impacts, and progressively worsens [13,14,15]. Therefore, the TBI model is a useful MCI research tool because it is simple, progressive, reproducible, and the severity of cognitive decline is relative to the number of impacts [16].

Chronic mild stress (CMS) is a behavioral model of depression that is caused by sequential exposure to variable mild stressors. CMS is characterized by anhedonia that may be reversed by chronic treatment with antidepressants [17]. However, most depression models show cognitive decline. Also, in the CMS animal model, mild cognitive deficit is accompanied, and antidepressants improve cognitive function in this model [18]. These memory deficits were related to the increased phosphorylation of Tau and APP processing, and the application of stress to wild type mice was suggested as an animal model of sporadic AD (Alzheimer’s disease) [19].

Currently, there are many methods on trial to prevent the progression of MCI to dementia. In particular, nonpharmacological methods, such as cognitive leisure activities, education and exercise, and pharmacological methods, such as vitamin E, donepezil, and intranasal insulin, have been tried [20,21,22,23]. However, more research is needed to prove their efficacy.

*Angelica gigas* (AG) has been used in traditional medicine to improve circulation, physical weakness, headache, dizziness, joint pain, abdominal pain, constipation, irregular menstruation, and bruises, etc. [24]. Known bioactive components of AG include decursin, decursinol angelate, and nodakenin [24]. It has been shown to improve liver function in rats treated with long-term ethanol. AG also lowered LDL cholesterol and inhibited nicotine sensitization in rats [25,26]. Finally, AG attenuated acetylcholinesterase activity and was neuroprotective against beta-amyloid peptide-induced memory impairment [27]. According to these findings, the AG extract was tested in TBI and CMS models to evaluate whether it would improve memory impairment in MCI.

## 2. Results

### 2.1. AG Improved TBI- and CMS-Induced Spatial Learning Deficit 

The effect of AG on spatial learning and memory was measured using the Morris water maze (Figure 1). During the five-day training, TBI impaired spatial learning. Supplementation of AG improved the TBI-induced deficit in spatial learning (Figure 1A). CMS impaired spatial learning (control vs. CMS), which improved following AG supplementation (CMS vs. CMS + AG) (Figure 1B). The sucrose preference of the CMS mice gradually decreased over six weeks. This declining preference meant that the mice developed an anhedonia-like tendency. AG did not affect this anhedonia-like behavior (Appendix A). 

### 2.2. AG Improved Short-Term Working Memory

The effect of AG on short-term working memory was measured using the Y-maze test (Figure 2). TBI did not have a significant effect on the alternation behaviors. However, AG increased the alternation behavior in both TBI and normal mice (Figure 2A). Similarly, CMS did not affect the alternation behavior in CMS mice. In contrast, AG treatment increased the alternation behavior (Figure 2B)

### 2.3. AG Had No Effect on Object Recognition Memory

The effect of AG on recognition memory was measured in the novel object test (Figure 3). AG had no effect on normal mice. However, the recognition memory in TBI model mice declined with AG treatment. The absence of a before–after change in the AG treated TBI group meant that AG prevented the adverse effect of TBI (Figure 3A). There was no significant effect of CMS and AG treatment on the recognition memory (Figure 3B)

### 2.4. The Effect of AG on Fear Memory

The effect of AG on fear memory was measured using the fear conditioning paradigm (Figure 4). The acquisition of fear memory in the TBI mouse model was not statistically higher than that of the control animals. AG treatment lowered the acquisition of fear memory in the TBI model mice (Figure 4A). There was no significant effect of TBI and AG on the consolidated contextual and cued fear memory (Figure 4B,C). The acquisition of fear memory in the CMS mice was lower than that of controls. There was no difference in the fear acquisition after AG treatment in CMS mice (Figure 4D). Compared to the normal mice, the CMS, AG, and CMS + AG mouse groups had lower levels of consolidated contextual fear memory. However, there were no differences among the CMS, AG, and CMS + AG groups (Figure 4E). Compared to normal mice, the AG and CMS + AG groups had lower levels of consolidated cued fear memory. In addition, AG reduced the consolidated cued fear memory in the CMS group of mice (Figure 4F).

## 3. Discussion

We investigated whether AG extract could prevent the progression of cognitive decline or improve memory in MCI models. We found that AG improved spatial learning and working memory but suppressed fear memory in the TBI model. In the CMS model, AG improved spatial learning and suppressed the cued fear memory.

MCI is a condition that increases a patient’s risk of developing dementia. Therefore, it is important that MCI is diagnosed early in order to prevent or limit dementia development [4]. Methods to improve sleep disorders, depression, one’s social network, and physical exercise are considered for the treatment of MCI [28,29,30]. Pharmacological interventions, such as cholinesterase inhibitors, nonsteroidal anti-inflammatory drugs, estrogen replacement therapy, Gingko biloba, and vitamin E, have not shown to prevent MCI progression to dementia [4].

The roots of AG have been used in traditional medicine to improve blood flow and anemia. They have also been used for their analgesic properties [31]. AG improves spatial memory, avoidance memory, and working memory in dementia models [27]. Among the components of AG, decursinol showed the highest inhibitory activity toward acetylcholinesterase [32]. Therefore, AG is expected to improve cognitive impairment in animal models.

In order to develop a valid MCI animal model, there must be subtle memory impairment [5]. The presence of depression-like symptoms may also be an important factor in the MCI model. Dementia and depression are known to have many associations. For instance, 60% of MCI cases that progress to AD are accompanied by depression [33].

The defect in a TBI model varies depending on the hitting area and velocity [14]. In contrast with severe TBI, mild TBI has minimal histological changes but apparent cognitive and emotional problems [8]. This association suggests that mild TBI can be used as an MCI model [34]. Such spatial memory deficit occurred in our TBI model mice, but other reflexes (such as the paw withdrawal reflex, righting reflex, and corneal reflex) were maintained (data not shown). Also, TBI is commonly known to cause disturbances in working memory. TBI in the parietotemporal regions has been shown to cause working memory deficits in mice [35]. However, working memory was normal in the mild TBI model made by repeated frontal impact; therefore, working memory is not always compromised in TBI [36]. Our measurements of working memory did not differ in the TBI model. Another common problem in TBI is the inability to recognize the source of information, such as facial recognition [37]. Face recognition memory corresponds to animal object recognition memory. There was a deficit in animal object recognition memory in the TBI models [38]. The TBI mice in this study also had decreased object recognition memory. These model mice also showed heightened fear memory; however, the results are controversial. One group found that there was no difference between normal animals and a mild repeated frontal TBI model with regard to conditioned fear after a severe CCI impact to the left parietal cortex [36]. However, a single impact above the skull increased anxiety and contextual fear in rats [39]. In mice, hippocampal-dependent fear memory decreased, but cued-dependent fear memory was not affected by TBI [40]. In our results, there was an enhanced acquisition of contextual fear in the TBI mice. However, after 24 h, the consolidated level of contextual and cued memory was not different with control animals. 

Stress has a variety of effects on cognitive function [41]. CMS is a depression model also characterized by a decrease in cognitive function associated with neuroimmune, neuroendocrine, and neurogenesis functions [4]. In this experiment, the CMS animals displayed anhedonia-like behaviors in the sucrose preference test and deficits in spatial memory. In other studies using CMS models, working memory was reduced in rats but was maintained in mice [42,43]. Our study similarly found that there was no working memory deficit in CMS mice. Reduction in object recognition memory in mice and increased contextual fear in rats were reported previously [42,44]. In the present study, CMS did not affect object recognition memory, but fear acquisition and contextual consolidation were reduced. These findings were inconsistent; however, rats that were exposed to social instability stress (daily 1-h isolation, change of cage partners) showed deficits in contextual and cued memory [45,46]. Taken together, in the TBI mice, spatial learning and object recognition memory were degraded. However, spatial memory, working memory, and fear memory were intact. In the CMS model, spatial memory was degraded, but working and object recognition memory were not affected, and cued fear memory was reduced. 

AG has known antibacterial, immune-stimulating, antiplatelet aggregation, neuroprotective, anti-inflammatory, and antioxidant properties [31]. We found that AG did not improve spatial memory in normal mice. However, it did improve memory in TBI and CMS animals. In addition, AG treatment led to improved working memory in normal mice of the TBI cohort. AG did not increase the recognition memory of the normal mice. However, recognition memory was only abnormal in the TBI mice at baseline. In the TBI and AG co-administration group, recognition memory was maintained. This result suggests that AG can prevent TBI-induced deficit. In the CMS cohort, however, the object memory of the CMS group was not affected. Therefore, AG also did not improve object memory in either the normal or the CMS mice. With regard to fear memory, the contextual and cued consolidated fear memory of normal and CMS mice were reduced in the CMS cohort. However, consolidated fear levels were not affected in the TBI cohort. Among AG’s known effects, its cholinesterase inhibition, improved blood flow, and anti-inflammatory properties may prevent cognitive decline and improve memory in these mouse models [31,32].

In conclusion, AG prevented the deterioration of spatial learning and object recognition memory in a mouse TBI model. AG also prevented the deterioration of spatial learning in the CMS model mice and improved working memory in normal mice. As TBI is a cognitive impairment gradually progresses, and chronic stress can cause AD-like pathologies, these findings suggest that AG may also prevent progressive cognitive decline in MCI animal models, which may be worth further research.

## 4. Materials and Methods 

### 4.1. Preparation of AG Extract

The dried AG root was purchased from the Junbu, Bonghwa, and Jecheon area and authenticated by professor Hui Jin. A voucher specimen was deposited in the Myongji Bioefficiency Research Center, Myongji University. AG was immersed in 70% ethanol and boiled for 4 h at 90 °C, 2 times. Then, filtered extracts were concentrated up to 25 Brix at 60 °C by depressurized evaporation, and then stored at 4 °C before using. The crude extract yield was 42.1% (*w*/*w*). The extract was dissolved in distilled water for administration into animals. HPLC was used to examine whether extracts contained nodakenin and decursin (Appendix A).

### 4.2. Animals and Experimental Groups

Seven-week-old male C57Bl/6 mice were purchased from Central Laboratory animals Inc. (Seoul, Korea) (*n* = 32 for cohort 1, *n* = 40 for cohort 2). The mice were housed under constant temperature and humidity with 12-h cycles of light/darkness. They had *ad libitum* access to rodent chow and water. After a week of habituation, the mice were randomly assigned into experimental groups. The mice in the TBI cohort were assigned to the following assignments: (1) control (*n* = 6): DW p.o. + no TBI; (2) AG (*n* = 6): *Angelica gigas* 1 mg/kg p.o + no TBI; (3) TBI (*n* = 8): DW p.o. + TBI; (4) TBI + AG (*n* = 12): *Angelica gigas* 1 mg/kg p.o. + TBI. The mice in the CMS cohort were assigned to the following assignments: (1) control (*n* = 10): DW p.o. + no CMS; (2) AG (*n* = 10): *Angelica gigas* 1 mg/kg p.o + no CMS; 3) CMS (*n* = 10): DW p.o. + CMS; 4) CMS + AG (*n* = 10): *Angelica gigas* 1 mg/kg p.o. + CMS. The outline of the experimental procedures is schematically represented in Figure 5. Animal studies were conducted in accordance with the Guide for the Care and Use of Laboratory Animals by the NIH. The protocols were approved by the Institutional Animal Care and Use Committee of Kyung Hee University (KHUASP(GC)-17-024).

### 4.3. Creating Animal Models

#### 4.3.1. TBI Model

A Cortical Contusion Injury device, which was purchased from Custom Design and Fabrication, Inc., (model eCCI-6.3), was used to simulate TBI. The mice were anesthetized with 3% isoflurane during the procedure. In order to induce TBI, a 3 mm rod impacted the scalp over the right hippocampal region at the velocity of 4 m/second and depth of 0.5 mm. This was repeated five times, with two days between each impact. Animals in the control and AG groups also underwent isoflurane anesthesia, but no impact.

#### 4.3.2. CMS Model

The stress procedure lasted for six weeks. The mice were randomly subjected to two types of stressors each day. The potential stressors included 2 h of immobilization, strobe light exposure, white noise exposure, cat urine exposure, and overnight food deprivation, water deprivation, or light exposure. During the weekend, the mice were exposed to wet bedding or a tilted cage for 24 h. The control mice were left undisturbed in their home cage.

### 4.4. Behavioral Tests

#### 4.4.1. Sucrose Preference

Anhedonia-like changes in the CMS mice were monitored using the sucrose preference test, which was performed at the 2nd, 4th, and 6th week. Consumption of water and sucrose by a single caged mouse was measured from 6 p.m. to 9 a.m. The preference index was calculated as follows: [sucrose consumption/(DW consumption + sucrose consumption)].

#### 4.4.2. Morris Water Maze

The water tank (200 cm diameter) was filled 0.5 cm above the platform with tap water and made opaque by white paint. For 5 days, the mice were trained to locate the submerged platform within 60 s according to the symbols placed around the walls. If the mouse was unable to escape within 60 s, it was guided to the platform. Each day of training included 4 sessions, each of which started in a different quadrant.

#### 4.4.3. Y-Maze 

The maze consisted of three corridors that were joined at the center at equal angles. After placing a mouse in the maze, the movements were recorded for five minutes. An alternation was defined as a sequential visit to each arm without the repetition of either of the two previous arms. The percent alternation was calculated as the number of correct alternations per total arm visits minus 2.

#### 4.4.4. Novel Object Test

In a plexiglas box (50 × 50 × 40 cm), two cylinder-shaped tin cans were introduced in two corners, 30 cm apart from each other. The mice were allowed to explore each object for 5 min. On the next day, the object of the right corner was replaced with a box-shaped novel object. The time that the mice spent exploring each object was recorded during the 5-min period. The animals were considered to be exploring an object when they were facing or sniffing the object. The recognition index was calculated by the ratio of time spent exploring the novel object over the total time spent exploring both objects.

#### 4.4.5. Fear Conditioning 

The fear conditioning test consisted of acquisition, contextual consolidation, and cued consolidation phases. For the acquisition, the mice were placed in the Passive/Active avoidance system chamber from Scitech Korea (model No. PAAS) and left undisturbed for two minutes. The subsequent sessions comprised a conditioned stimulus (2000 Hz tone, 30 s) that co-terminated with an unconditioned stimulus (electric foot shock, 0.45 mA, 2 s), and intertrial intervals (30 s) were repeated four times. The mice were left in the chamber for an additional two minutes. Freezing was measured during the first two minutes and last two minutes of the 4 intertrial intervals. On the following day, the animals were placed in the chamber for five minutes. Freezing was measured as a contextual fear memory. On the third day, the mice were placed in a novel chamber for three minutes, followed by three minutes of exposure to a 2000 Hz tone. Freezing was measured before and during the tone exposure. 

### 4.5. Statistical Analysis

Analysis of variance (ANOVA) and repeated measures ANOVA, following Tukey’s HSD post-hoc test, were performed using SPSS 23 (IBM). *p*-Values < 0.05 were considered significantly different. The normality of distribution of variables was tested by the Shapiro-Wilk test.

## Figures and Tables

**Figure 1 nutrients-12-00097-f001:**
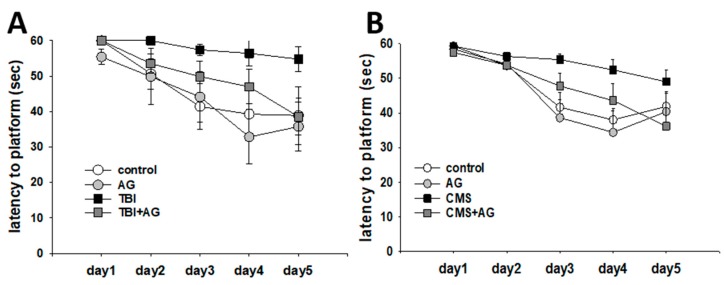
Spatial learning by *Angelica gigas* (AG) in traumatic brain injury (TBI) and chronic mild stress (CMS) mice as measured by the latency to the platform in the Morris water maze (MWM). (**A**) TBI model. There were repetitive training (within) effects [F(4,104) = 13.9, *p* < 0.001] and treatment group (between) effects [F(3,26) = 2.27, *p* = 0.044], but with non-significant within–between interaction differences [F(12,104) = 1.1, *p* = 0.36]. The post-hoc pairwise comparison showed a difference between control vs. TBI (*p* = 0.033), TBI vs. TBI + AG (*p* = 0.041). (**B**) CMS model. There were repetitive training effects [F(4,196) = 31.3, *p* < 0.001], treatment group effects [F(3,49) = 3.1, *p* = 0.034], and within–between interactions [F(12,196) = 2.2, *p* = 0.012]. Post-hoc pairwise comparison revealed a difference between the control and CMS groups (*p* = 0.019), AG vs. CMS (*p* = 0.01), CMS vs. CMS + AG (*p* = 0.042). All data were normally distributed and are represented as means ± S.E.M. Control: vehicle (DW) treated; AG: *Angelica gigas* 1 mg/kg; TBI: vehicle treated + traumatic brain injury; TBI + AG: *Angelica gigas* 1 mg/kg + traumatic brain injury; CMS: vehicle treated + chronic mild stress; CMS + AG: *Angelica gigas* 1 mg/kg + chronic mild stress. Repeated measure ANOVA, Tukey’s HSD post-hoc test.

**Figure 2 nutrients-12-00097-f002:**
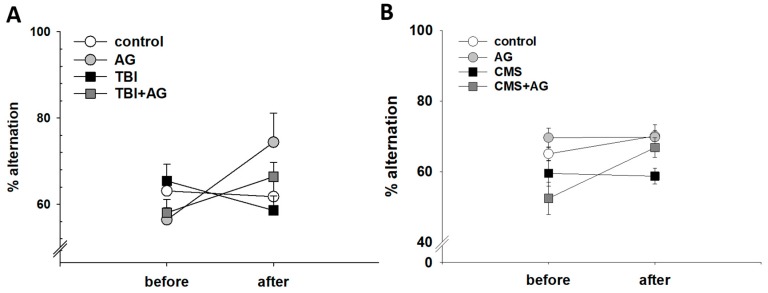
Short-term working memory by AG in TBI and CMS mice, measured by the percent alternation in the Y-maze. (**A**) TBI model. There were before–after (within) effects [F(1,28) = 3.57, *p* = 0.012], treatment group (between) effects [F(3,28) = 3.23, *p* = 0.081], and within–between interactions [F(3,28) = 3.28, *p* = 0.036]. There were no before–after changes in the control (*p* = 0.6) and TBI (*p* = 0.38), but improvements in the AG (*p* = 0.008) and TBI + AG (*p* = 0.047). (**B**) CMS model. There were before–after effects [F(1,36) = 6.2, *p* = 0.018] and treatment group effects [F(3,36) = 11.8, *p* < 0.001], but no significant within–between interaction differences [F(3,36) = 1.5, *p* = 0.24]. There was no before–after change in the control group (*p* = 031), AG (*p* = 0.97) or CMS (*p* = 0.33) groups. However, there was an increase of % alternation in the CMS + AG group (*p* = 0.006). All data were normally distributed and are represented as means ± S.E.M. Control: vehicle (DW) treated; AG: *Angelica gigas* 1 mg/kg; TBI: vehicle treated + traumatic brain injury; TBI+AG: *Angelica gigas* 1 mg/kg + traumatic brain injury; CMS: vehicle treated + chronic mild stress; CMS+AG: *Angelica gigas* 1 mg/kg + chronic mild stress. Repeated measures ANOVA, Tukey’s HSD post-hoc test.

**Figure 3 nutrients-12-00097-f003:**
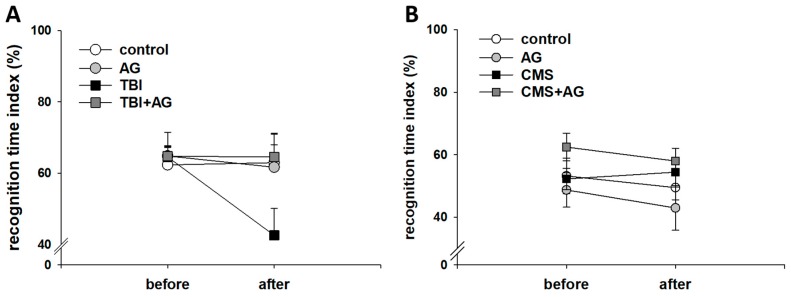
Object recognition memory by AG in the TBI and CMS mice as measured by the recognition time index in the novel object test. (**A**) TBI model. There were before–after (within) effects [F(1,28) = 8.3, *p* = 0.008], treatment group (between) effects [F(3,28) = 0.67, *p* = 0.58], and within–between interaction effects [F(3,28) = 3.6, *p* = 0.026]. There was no before–after change in the control (*p* = 0.93), AG (*p* = 0.59) and TBI + AG (*p* = 0.97) groups, but there were decreases in the TBI (*p* = 0.001) group. (**B**) CMS model. There were no before-after effects [F(1,36) = 0.32, *p* = 0.57], treatment group effects [F(3,36) = 1.2, *p* = 0.31], or within–between interactions [F(3,36) = 0.12, *p* = 0.95]. All data were normally distributed and are represented as means ± S.E.M. Control: vehicle (DW) treated; AG: *Angelica gigas* 1 mg/kg; TBI: vehicle treated + traumatic brain injury; TBI + AG: *Angelica gigas* 1 mg/kg + traumatic brain injury; CMS: vehicle treated + chronic mild stress; CMS+AG: *Angelica gigas* 1 mg/kg + chronic mild stress. Repeated measures ANOVA, Tukey’s HSD post-hoc test.

**Figure 4 nutrients-12-00097-f004:**
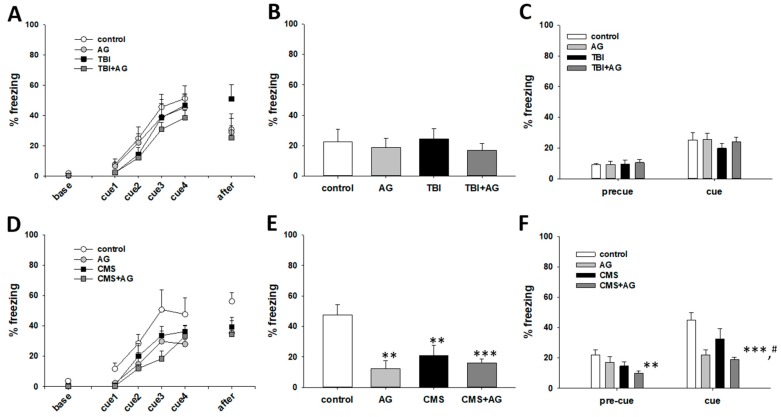
Fear memory by AG in TBI and CMS mice, as measured by the freezing time in the fear conditioning test. (**A**) Fear acquisition in the TBI model. There were time (within) effects [F(5,140) = 45.7, *p* < 0.001], treatment group (between) effects [F(3,28) = 3.25, *p* = 0.03], but no within–between interaction differences [F(15,140) = 1.1, *p* = 0.36]. There were no differences between the treatment groups at base, cue1, cue2, cue3, or cue4. However, the differences between the TBI and TBI + AG groups were significant (*p* = 0.039). In addition, the differences between the TBI and control groups (*p* = 0.16), and TBI vs. AG (*p* = 0.13) groups were not significant. (**B**) Consolidated contextual fear memory in TBI model. There were no group differences in the freezing response to the context [F(3,28) = 0.36, *p* = 0.78). (**C**) Consolidated cued fear in the TBI model. There were procedure (within) effects [F(1, 28) = 61.6, *p* < 0.001], no treatment group (between) effects [F(3,28) = 0.29, *p* = 0.83], or within–between interactions [F(3,28) = 0.64, *p* = 0.59]. There was a significant increase in the freezing comparing cue vs. precue in the control (*p* < 0.001), AG (*p* < 0.001), TBI (*p* = 0.005), and TBI + AG groups (*p* < 0.001). (**D**) Fear acquisition in the CMS model. There were time (within) effects [F(5,95) = 27.8, *p* < 0.001], treatment group (between) effects [F(3,19) = 4.9, *p* = 0.011], but no within–between interaction effects [F(15,95) = 0.47, *p* = 0.95]. Fear acquisition in AG (*p* = 0.021), CMS (*p* = 0.035), and CMS + AG (*p* = 0.001) were lower than controls throughout the fear acquisition procedure. There were no statistical differences among the AG, CMS, and CMS + AG groups. (**E**) Consolidated contextual fear memory in the CMS model. There were group differences in the freezing response to the context [F(3,19) = 9.8, *p* < 0.001). The contextual fear in AG (*p* = 0.002), CMS (*p* = 0.008), and CMS + AG (*p* < 0.001) was lower than those of the controls. (**F**) Consolidated cued fear in the CMS model. There were procedure (within) effects [F(1,19) = 54.7, *p* < 0.001], treatment group (between) effects [F(3,19) = 9.3, *p* = 0.001], and within–between interaction differences [F(3,19) = 4.8, *p* = 0.012]. There was a significant increase in freezing between cue and precue in the control (*p* < 0.001), CMS (*p* < 0.001), and CMS + AG (*p* = 0.002) groups, but not in the AG group (*p* = 0.31). At pre-cue, the control vs. CMS + AG (*p* = 0.001) groups were statistically different. At cue, the control vs. AG, (*p* = 0.004); control vs. CMS + AG, (*p* < 0.001); CMS vs. CMS + AG, (*p* = 0.017) were all statistically different. All data were normally distributed and are represented as means ± S.E.M. Control: vehicle (DW) treated; AG: *Angelica gigas* 1 mg/kg; TBI: vehicle treated + traumatic brain injury; TBI + AG: *Angelica gigas* 1 mg/kg + traumatic brain injury; CMS: vehicle treated + chronic mild stress; CMS + AG: *Angelica gigas* 1 mg/kg + chronic mild stress. ** *p* < 0.01, *** *p* < 0.001 vs. control. ^#^
*p* < 0.05 vs. CMS. ANOVA and repeated measures ANOVA, Tukey’s HSD post-hoc test.

**Figure 5 nutrients-12-00097-f005:**
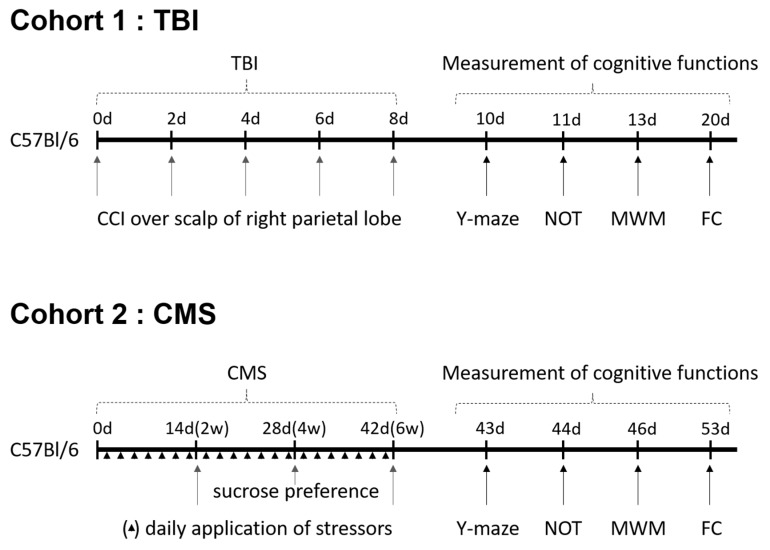
Experimental outline of mild cognitive impairment mice models induced by TBI (Cohort 1) and CMS (Cohort 2). TBI: Traumatic brain injury, CCI: Controlled cortical impact, Y-maze: Y shaped maze test, NOT: Novel object test, MWM: Morris water maze, FC: Fear conditioning, CMS: Chronic mild stress.

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
