# Peer review of "Evaluating the Memory Enhancing Effects of Angelica gigas in Mouse Models of Mild Cognitive Impairments"

_nutrients, 2019, doi:10.3390/nu12010097_

Round 1

Reviewer 1 Report

I have some methodological concerns regarding this manuscript:

The investigational drug , Angelica gigas, is an ethanol extract of a plant route.  There is no indication of what the active component might be, and what 'contaminants' there are.  The vehicle control appears to be distilled water (DW).  Distilled water is not an appropriate control for a drug administered in an ethanol diluent. 

The is no indication, within the description of the novel object recognition test, whether the introduction of the novel object was always to the same side or whether it was alternated.

Within the data analysis section, there is no discussion as to whether the derived data were normally distributed and whether ANOVA and t-tests were appropriate tests for the data.

Finally, the introduction fails to convince me that traumatic brain injury and chronic mild stress are appropriate models of MCI or early dementia.  The findings indicate that Angelica gigas may overcome the adverse effects of traumatic brain injury and chronic mild stress, but this does not mean that it will also alter the progression of dementia 

Author Response

I have some methodological concerns regarding this manuscript:

The investigational drug , Angelica gigas, is an ethanol extract of a plant route.  There is no indication of what the active component might be, and what 'contaminants' there are.  The vehicle control appears to be distilled water (DW).  Distilled water is not an appropriate control for a drug administered in an ethanol diluent. 

Thank you for your comments. All of the modifications we made are highlighted in yellow.First, Known active compounds of Angelica gigas are decursin, decursinol, nodekenin and etc. Among these, decursin is the most abundant compound and we measured the amount of decursin and nodakenin using the HPLC. This result is included to the manuscript as supplementary figure2. Also, active compounds of AG were included in the introduction section.

Second, about the vehicle. Ethanol was used in the process of extracting AG and later vaporized entirely and no longer remained in the extract. Later, when feeding to animals, the extract was dissolved in distilled water.

The is no indication, within the description of the novel object recognition test, whether the introduction of the novel object was always to the same side or whether it was alternated.

We replaced the object on the right with a new square-shape object. We specified this in text.

Within the data analysis section, there is no discussion as to whether the derived data were normally distributed and whether ANOVA and t-tests were appropriate tests for the data.

All of the measured data were tested for the normal distribution by Shapiro-Wilk test. We commented this in the method section and included in the figure legend.

Finally, the introduction fails to convince me that traumatic brain injury and chronic mild stress are appropriate models of MCI or early dementia.  The findings indicate that Angelica gigas may overcome the adverse effects of traumatic brain injury and chronic mild stress, but this does not mean that it will also alter the progression of dementia 

We suggested TBI and CMS as a MCI model because, cognitive decline worsens progressively in TBI patients and in TBI models. Also stress in suggested as one of the aggravating factors of dementia, and in CMS animal models, Alzheimer-like pathological findings such as APP processing and increase in p-Tau occurs. This can mean that the more stress applied, the faster the progression to dementia. We agree on your point that the data provided does not mean AG will alter the progression of dementia. To find out, the AG should have been administered after the model was completed, but it had to be tested first to see if it worked. However, as TBI is a model of progressive cognitive decline, and the intensity and duration of stress exacerbates the cognitive impairment, we suggested the improvement of cognitive impairment by AG would also prevent the progression of dementia.

Also, the title does not use the wording that AG prevents MCI from proceeding.

Reviewer 2 Report

In this study, TBI and CMS mouse models were used to show the effect of Angelica gigas extract on a number of neurological parameters. It is an interesting study, although not completely new since the authors described that effects on memory have been shown before. Nevertheless it is interesting evidence.

In general, the text should be revised by a native English speaking person. Some sentences are not complete or grammatically correct. I already indicated some changes in the attached document.

I believe the Results can be more compact since you use nice figures that tell the story.

Furthermore, the last sentence of the conclusion should be revised. You only can hypothesize what these results mean for progression towards dementia.

Author Response

In this study, TBI and CMS mouse models were used to show the effect of Angelica gigas extract on a number of neurological parameters. It is an interesting study, although not completely new since the authors described that effects on memory have been shown before. Nevertheless it is interesting evidence.

We thank you for your detail review on our manuscript. All of the modifications we made are highlighted in yellow. The difference from previous studies is that we looked into more detailed cognitive aspects such as spatial memory, working memory, object recognition memory and fear memory. Also, the animal models were more close to an MCI condition than AD.

In general, the text should be revised by a native English speaking person. Some sentences are not complete or grammatically correct. I already indicated some changes in the attached document.

This manuscript was checked by a professional editing company (eWorldEditing). Later on we made some minor modifications in the abstract and discussion part. We made all of the corrections suggested by you and some more corrections on our own. If you think the sentences still needs to be checked, we will request an editing service from another company.  A certificate is uploaded as a PDF file.

I believe the Results can be more compact since you use nice figures that tell the story.

As you suggested, we made the result more compact.

Furthermore, the last sentence of the conclusion should be revised. You only can hypothesize what these results mean for progression towards dementia.

Thank you. We agree that is an appropriate suggestion. We revised the conclusion by taming the expression

Round 2

Reviewer 1 Report

All of my comments and queries have been addressed.